# The Controversial Issue of Hypervitaminosis B12 as Prognostic Factor of Mortality: Global Lessons from a Systematic Review and Meta-Analysis

**DOI:** 10.3390/nu17132184

**Published:** 2025-06-30

**Authors:** Edith Valdez-Martínez, Horacio Márquez-González, Ricardo Ramírez-Aldana, Miguel Bedolla

**Affiliations:** 1Health Research Council, Mexican Institute of Social Security, Centro Médico Nacional SXXI, Avenida Cuauhtémoc 330, Colonia Doctores, Mexico City 06720, Mexico; 2Heart Disease Department, Cardiology Hospital, Centro Médico Nacional SXXI, Mexican Institute of Social Security, Avenida Cuauhtémoc 330, Colonia Doctores, Alcaldía Cuauhtémoc, Mexico City 06720, Mexico; horaciohimfg@gmail.com; 3Department of Mathematics, Faculty of Sciences, National Autonomous University of Mexico, Circuito Exterior, Ciudad Universitaria, Mexico City 04510, Mexico; ricardoramirezaldana@ciencias.unam.mx; 4Independent Researcher, 3311 Rock Creek Run, San Antonio, TX 788230, USA; miguel.bedolla2@gmail.com

**Keywords:** vitamin B12, mortality, prognosis, meta-analysis

## Abstract

Objective: To test whether hypervitaminosis B12 is useful for prognosis of all-cause mortality. Methods: Meta-analysis of longitudinal, observational, epidemiologic studies. PubMed, Scopus, Web of Science, Google Scholar, and ProQuest One Academic were searched from inception to 30 June 2024. Studies including humans aged ≥18 years with hypervitaminosis B12, and in whom the outcome variable was all-cause mortality, were included. Two reviewers screened, abstracted (using standardized data collection sheet), and appraised articles (ROBINS-E framework) independently. Frequentist and Bayesian approaches were used for the meta-analysis. Results: A total of 28 studies were included in the meta-analysis (among the 69,610 participants, 15,815 all-cause deaths were reported). High serum levels of B12 increased marginally the risk of all-cause mortality specifically among chronic diseases (RR = 1.40; 95% IC = 1.05 to 1.85) and hospitalized (RR = 1.57; 95% IC = 1.19 to 2.07). In the meta-regression, these results were not statistically significant. The Bayesian analysis confirmed the risks of the mentioned groups; however, it was limited by the number of studies that contained the necessary information. The methodology applied and the clinical heterogeneity of each included study bring up the idea that artefacts might be involved when mortality is found to be high for hypervitaminosis B12. Conclusions: This meta-analysis did not show that hypervitaminosis B12 represents a higher risk of all-cause mortality in adults.

## 1. Introduction

Hypervitaminosis B12 is the short term that will be used to name high serum concentrations of vB12 (vitamin B12, cobalamin).

The absorption and transport of vB12 entail a highly integrated multiprotein network [1,2]. Worldwide research indicates that vB12 is involved in various cell signaling pathways, including isolation of nerve axons, DNA replication and transcription, mitochondrial functionality, and metabolism of various amino acids [1,2]. In this manner, the occurrence of hypervitaminosis B12 may depend on a variety of risk factors likely to result of alterations in its metabolism, its blood transport, and its excretion (urine and feces), irrespective of the causes that may generate it. Throughout the years, there has been quantitative evidence that associates hypervitaminosis B12 with the risk of all-cause mortality (in addition to associating it with the gloomy clinical course of several conditions) [3,4,5,6,7,8,9,10,11,12,13,14,15,16,17], but there is also quantitative evidence that denies such associations [18,19,20,21,22,23,24,25,26,27,28,29,30]. In most of these primary studies the most frequent error is a causal interpretation of predictor effects.

Findings of a 2024 dose-response meta-analysis of 22 studies [31] shows that for each 135 pg/mL increase in serum vB12 concentration, there is a 4% higher risk of all-cause mortality in the general population (aHR 1.04, 95% CI 1.01 to 1.08; *n* = 8; p non-linearity = 0.11) and a 6% higher risk for all-cause mortality in older adults (aHR 1.06, 95% CI 1.01 to 1.13; *n* = 4; p non-linearity = 0.78). It also shows that a B12 concentration > 813 pg/mL is associated with all-cause mortality risk (aHR 1.50, 95% CI 1.29 to 1.74; *n* = 10; *p* < 0.01). Nevertheless, the findings/conclusions are not supported by data/study evidence; that is, the building blocks of analysis and interpretation are not completely available, raising concerns about their credibility and reliability, either related to the reviewing the evidence or the process of statistical analyzing of the predictor effects or the synthesis of information (internal validity).

A 2024 systematic review [32] that aims to provide knowledge support on the association of hypervitaminosis B12 and all-cause mortality in adults finds that this potential association is constrained by limitations inherent in the studies included, remaining highly uncertain. The aggregative synthesis of this review is based on six longitudinal analytic observational studies encompassing 94,313 human subjects with all-cause deaths.

Scoping of the worldwide literature (primary studies), over hypervitaminosis B12, showed that studies are based on the frequentist or classic statistical inference.

In this meta-analysis, quantitative data were pooled for analysis and modelling using frequentist statistical analysis and Bayesian statistical methods. Each of the two approaches made a unique contribution by focusing not just on finding out to what extent hypervitaminosis B12 can predict or improve the prediction of all-cause mortality risk in adults, but further on the question of how much hypervitaminosis B12 should change the assertion about its role as a prognostic marker. The purpose of this meta-analysis is therefore to provide decision support. This meta-analysis is both timely and relevant, and the evidence is urgently needed to better understand the support that clinicians require in addressing hypervitaminosis B12.

The following objective was defined to help organize the search and analysis of evidence: to test whether hypervitaminosis B12 is useful in a prediction model for all-cause mortality risk in adults, and to establish the support for different values of the effect of hypervitaminosis B12 to arrive at a judgment about its effect on mortality.

## 2. Materials and Methods

The meta-analysis protocol was registered in PROSPERO (registration number CRD42022361655). The review methods are briefly described below, and more details are presented in the online Appendix A. This study was performed according to the Preferred Reporting Items for Systematic Reviews and Meta-analyses (the PRISMA statement). The review involved skilled researchers with different areas of expertise: two clinicians (EV, HM), one librarian (JAF), and one mathematician (RR). Throughout the research process, the research team held many joint sessions, the analysis process required that each study to be read repeatedly to ensure that all concepts and statistical information were understood and integrated and the relationships between each study were explored.

### 2.1. Inclusion Criteria

Longitudinal, observational studies carried out in clinical or epidemiological scenarios, in humans aged ≥18 years with high serum levels of vB12 and in which the outcome variable was all-cause mortality, as well as adjusted HR values were reported. Publications included those written in the English and Spanish language.

### 2.2. Exclusion Criteria

Studies with very low methodological quality or with insufficient data about mortality or levels of vB12 or when study subjects were not clearly described were excluded.

### 2.3. Information Sources and Search Strategy

PICO [33] was the framework used to locate studies using the following Mesh terms: “adult” OR “old*”; AND “vitamin B12” OR “cyanocobalamin” AND “high” OR “elevated”; AND “mortality” OR “death” OR “survival”; AND prognosis. Additionally, terms that had the same or nearly the same meaning as these were incorporated.

PubMed, Scopus, Web of Science, and Google Scholar were searched electronically. ProQuest One Academic and Google were utilized to identify ongoing and unpublished primary studies. All these searches were supplemented with hand checking of reference lists. The last search was conducted on 30 June 2024. EndNote software^TM^ 21 helped us organize and manage the reference workflow.

### 2.4. Selection Process

One researcher (JAF) performed searches in the electronic databases to identify the primary studies of possible interest. Titles and abstracts were reviewed by JAF and checked by EV. These same two researchers (EV, JAF) checked the reference lists of studies from search results to identify further potentially eligible studies.

To decide whether a study should be assessed in full, two researchers (EV, HM) independently screened the abstract of every record retrieved. These two researchers (EV, HM) also independently assessed the full text of all potentially relevant studies. Any queries were resolved by consensus. One researcher (EV) listed the excluded studies with the reason for exclusion.

### 2.5. Data Extraction

Two researchers (EV, HM) performed the data extraction independently, using standardized data collection sheet adapted from National Institute for Health and Care Excellence Guideline [34] to obtain the following information: (a) characteristics of the study (name of the first author, country and publication year); (b) Methods and design (objective and type of study, setting and sociodemographic of study population, eligibility criteria, inception cohort and indicators of disease course and severity, sample and sampling, follow-up time, and attrition rate), as well as a list of co-variables selected based on ability to predict mortality, fitting for a prediction study, e.g., cardiovascular disease, cancer, diabetes, liver and kidney failure, in hospital, outpatients, elderly; (c) Laboratory measurements of vB12 (applied measurement techniques, baseline and times in which vB12 was quantified, cut-off points, measurement units). All measurements reported as pmol/L, ng/L, were converted to pg/mL (d) Operational definition of mortality and survival; (e) Statistical analysis used to estimate the predictive value of hypervitaminosis B12 on mortality.

### 2.6. Search Outcome

The initial electronic searches identified 2986 citations (Figure 1). The reference lists’ screening identified 23 further research studies. From these references, 96 required a full-document screen to determine if they met the inclusion criteria. Out of 30 studies that met the inclusion criteria, 2 studies were excluded because they had a fatal flaw (the threshold for exclusion); thus, 28 studies were included in the statistical pooling.

### 2.7. Quality Assessment

Two researchers (EV, HM), using a standardized data collection sheet, independently rated the quality of each study using the ROBINS-E framework (Risk Of Bias In Non-randomized Studies-of Exposure) [35]. Any disagreements were resolved through discussion between the two appraisers. Two studies had a fatal flaw (the threshold for exclusion) [Appendix A]. The fatal flaws that render their findings questionable are the lack of internal validity due of critical defects in the methodological execution and analysis or, more precisely, (i) flawed data collection, i.e., there is no calculation of the sample size and of the estimation of the statistical power of the sample included. It is also not clear what is the sampling frame; (ii) Flawed methodological execution; specifically, relevant confounding variables are missing, and the statistical approach does not contribute to the classification of the prediction process; (iii) Limited practical relevance/implications due to results/conclusions not actionable.

### 2.8. Statistical Analysis

First, a meta-analysis was applied considering as intervention effect the standardized difference of means (SDM), of vB12 levels, between death and live individuals.

After that, it was considered the relative risk (RR) of dying, between individuals with high and normal vB12 levels, as intervention effect. A subgroup analysis was performed using a common variance between study groups with an inverse variance estimation method [36]. The groups considered were disease type and hospitalization. Tests to determine differences between and within groups were obtained, as well as the estimated RR in each group. To determine small-study effects, a funnel plot for all studies and by group were derived. Asymmetry or small-study effects tests by group were estimated. For both, the analysis based on SDM and RR, fixed and random-effects models were fitted. All inference was presented for only one model type, which was chosen according to a test of heterogeneity (Q test) and other heterogeneity measures: $I^{2}$ and the variance associated to the random effect or between studies variance, $\tau^{2}$.

Meta-regression models [37] were fitted, using a random-effects model and estimation based on a restricted maximum-likelihood (REML) estimator. As inputs, the same variables used for the group analysis plus a variable indicating whether a study was aimed or not to an elderly population. Both univariable and multivariable models were fitted. To identify whether random effects were required and whether all variables were jointly significant, tests of residual heterogeneity and of moderators were obtained, respectively. We also obtained Z-tests to determine significant inputs. Some regrouping of categories and deletion of studies was used.

Finally, network meta-analyses based on Bayesian methods [38] were fitted. We considered a likelihood based on a binomial distribution, a logit link function, and random effects. To obtain the stationary a posteriori distribution, four chains, 10,000 simulations, and a burn-in of 5000 were used. The thirds (T1, T2, T3) of vB12 levels as treatments or arms were considered. Since all studies that met inclusion criteria were studied, there was at least one direct comparison between them, all models have an associated complete graph (all nodes are joined by edges). The trace plots and Gelman–Rubin diagnostics were checked to determine whether the stationary distribution was reached. Rank probabilities on each model considering that lower values of the occurrence of the index event, ñin our case death, were preferred. Thus, a rank of 1 corresponded to a lower probability of dying and 3 to the highest. Statistical comparisons, with T1 as reference, were performed; as well as ranking analysis based on the SUCRA score. Global analysis, by disease type and by study setting (in-hospital-based studies or general population-based studies) were all performed. For all analyses, R version 4.2.1 and the following libraries: meta, metafor, rjags, gemtc, and dmetar, were used.

## 3. Results

### 3.1. Characteristics of Included Studies

Appendix A summarizes the 28 studies included in the meta-analysis, comprising 69,610 subjects and 15,815 all-cause of deaths. All these studies showed that different quantification techniques and methods to establish cut-off points were used to measure and to define hypervitaminosis B12. The studied samples were divided into two halves, thirds, quarters or fifths, the highest part of the data distribution was the one that appears associated with the risk of all-cause mortality.

The conclusions of all the 28 studies are based on a single vB12 measurement and a single multimorbidity and/or disease severity quantification in time?over years of follow-up. Roughly speaking, follow-up length varied between 0.9 and 132 months.

In 18 studies, the participating subjects were enrolled at or very soon after the vB12 quantification; and in three in a minimum interval of four weeks before or after enrolment, 7 studies did not specify the time of enrolment; thus, time of study enrolment varies with respect to exposure (serum vB12 concentrations) across the studies included. Overall, inclusion in the studies was considered as the baseline to analyze subsequent mortality.

Fifteen population-based studies were derived from national surveys [10,15,17,21,25,28,29,30] or surveys of residents in a defined geographical territory [7,11,16,20,24,26,27]. One institution-based cohort study [23], and twelve uni-centre studies (in-hospital-based studies and outpatient-based studies) [3,4,5,6,8,9,12,13,14,18,19,22].

### 3.2. Means Difference

Eight studies, with data on the living and the dead humans, were analyzed to determine whether there was a significant difference between SMD of vB12 levels when dead and live people were compared. Significant heterogeneity ($I^{2}=9$2.5%; variance estimated of 0.1694; Q = 93.63; *p* < 0.001) was observed; consequently, a random-effects model was used. The estimated SMD was 0.26 (95% CI of −0.05 to 0.57; *p* = 0.104). Hence, the difference was significant when we used a 0.11 significance level. The corresponding forest plot and funnel plot, in which a couple of studies seem to have publication bias, are shown in Appendix A.

### 3.3. Group Analyses

Twenty studies were used because they included all necessary information for the present analyses. Significant heterogeneity was observed ($I^{2}$ = 89.7%; variance estimated of 0.1529; Q = 184.51; *p* < 0.001). Thus, a random-effects model was used. The RR estimated was 1.25 (95% CI = 1.04 to 1.50), which indicated that the individuals with high B12 levels were 1.25 more likely to die than those with normal B12 levels.

Firstly, a model considering groups associated with the presence of different diseases was used (Figure 2). Utilizing random effects, the RR estimated did not significantly differ between groups (*p* = 0.353), though there was a significant difference between studies in each group (*p* < 0.001), observing a significant RR for chronic disease (1.40; 95% CI = 1.04 to 1.50). Each group was also analyzed separately, including a test to identify asymmetry in the funnel plot. Significant heterogeneity between cancer mortality studies (*p* < 0.001) and a not significant RR (*p* = 0.335) was observed. For cardiovascular mortality, there is neither heterogeneity between studies (*p* = 0.9021) nor significant RR (1.10; *p* = 0.078). For chronic disease mortality, there were a significant heterogeneity between studies (*p* < 0.001), a significant RR (1.41; *p* = 0.036) and significant asymmetry in the funnel plot (*p* = 0.0163). In general population-based studies, there was significant heterogeneity (*p* < 0.001), but neither significant RR (*p* = 0.658) nor significant asymmetry between studies (*p* = 0.8521).

Further on, a model considering groups associated with in-hospital-based studies versus outpatient-based studies was utilized (Figure 3). Using random effects, the RR showed a significant difference between groups (*p* = 0.021) and between the studies in each group (*p* < 0.001). Additionally, a significant RR (1.57; 95% CI = 1.19 to 2.07) for in-hospital-based studies was observed. When each group was analyzed separately, a significant heterogeneity between in-hospital-based studies (*p* < 0.001), a significant RR (1.59; *p* = 0.006), and a significant asymmetry in the funnel plot (*p* = 0.019), were observed. In general population-based studies, there was significant heterogeneity (*p* < 0.001); however, the RR (*p* = 0.405) and the asymmetry between studies (*p* = 0.0873) were not significant.

### 3.4. Meta-Regression

According to the results obtained in the grouping analysis, when univariable models were fitted, the variable associated with in-hospital-based studies was significant (*p* = 0.04), as expected, but studies by type of disease was not (z-tests for all dummy variables; *p* > 0.05). Additionally, the elderly variable was not significant (*p* = 0.84). Then, all the studies with chronic diseases (reported in their study objective) were grouped in a single category, called chronic disease. Here, four studies corresponding to those with the largest publication bias were eliminated from the analysis (Appendix A). After that, a meta-regression including all variables was fitted (Table 1). In this model, the variable associated with in-hospital-based studies was still significant at a significance level of 0.1 (*p* = 0.08) with an estimated coefficient of 0.58, and an exponentiated value of 1.316, indicating that after controlling for the other variables the risk of dying increased at higher vB12 levels. However, according to the test of moderators all variables were not simultaneously significant (*p* = 0.33).

### 3.5. Sensitivity Analyses

In consideration of the totality of the data as a whole and by subgroups, through a leave-out-one method, all the results remained, except for the chronic diseases group, which was shown to have statistical significance for all studies at a 0.1 significance level but not at a 0.05 significance level.

### 3.6. Networks

Since all models have an associated complete graph, the corresponding graphs are not presented. Additionally, since a Bayesian method was used, the trace plots associated with each parameter, in all models, were checked to ensure that each showed the random behavior expected for a stationary process. In the same line, the Gelman–Rubin diagnostic was, in all cases, almost of one.

In consideration of rank probabilities (Figure 4), for the global analysis, T1 had a rank of one (less probability of dying) in 66.74% of the simulations, whereas T2 and T3 had a higher percentage to be in rank 2 and 3, respectively. Thus, at higher vB12 levels, there was a higher probability of dying, as expected. However, when considering comparisons with T1 as reference, there was not a significant difference between T2 or T3 with T1, OR = 1.3 (95% credibility interval 0.41 to 4.8) and OR = 2.5 (95% credibility interval 0.83 to 9.7), respectively. When the SUCRA score was considered, T1 has the highest score (0.816), i.e., less possibility of dying, followed by T2 (0.603).

For general population-based studies, T1 had a rank of 3 (more probability of dying) in 70.61% of the simulations, whereas there was a higher percentage that belonged to rank 1 for T2, 87.60%. However, there was not a significant difference between T2 or T3 with T1 (third of reference), OR = 0.59 and OR = 0.86 (95% credibility intervals of 0.31 to 1.1; and 0.45 to 1.6), respectively. When the SUCRA score was considered, T2 had the highest score (0.923). For the group formed by studies concerning chronic diseases mortality, there were percentages of 84.24%, 72.03%, and 84.90% of having a rank of 1, 2, and 3 for T1, T2, and T3, respectively. That is, higher vB12 levels indicate a greater likelihood of dying. In this case, the comparison between T1 and T3 was significant, OR = 6 (95% credibility interval 1.01 to 50.4), whereas the comparison between T1 and T2 was not, OR = 2.4 (95% credibility interval 0.42 to 18.0). When the SUCRA score was considered, T1 has the highest score (0.914).

For in-hospital-based studies, T1 had a rank of 1 (less likelihood of dying) in 97.37% of the simulations, while there was a higher percentage to belong to rank 3 for T3, 97.80%. Additionally, when considering comparisons with T1, there was a significant difference between T1 and T3, OR = 16 (95% credibility interval 4.5 to 83), but no significant difference between T1 and T2, OR = 3.3 (95% credibility interval 1 to 19). When the SUCRA score was considered, T1 had the highest score (0.986).

For general population-based studies, there were percentages of 80.55% and 77.68% that had a rank of 3 and 1 for T1 and T2, respectively. That is, lower vB12 levels indicated a greater possibility of dying, whereas normal vB12 levels decreased it. In this case, neither the comparison between T1 and T2 nor the comparison between T1 and T3 were significant, OR = 0.68 and OR = 0.81 (95% credibility intervals 0.41 to 1.1, and 0.49 to 1.3), respectively. Finally, the SUCRA score had the highest score for T2, 0.872.

## 4. Discussion

This meta-analysis, which sampled 28 longitudinal-observational studies carried out in clinical or epidemiological scenarios, focused on testing whether hypervitaminosis B12 is useful in a prediction model for all-cause mortality risk in adults and establishing the different values of the effect of hypervitaminosis B12 that let the clinician to arrive at a judgment about its effect on mortality.

The findings of this meta-analysis indicate that hypervitaminosis B12 marginally increased the risk of all-cause mortality specifically among chronic diseases (RR = 1.40; 95% IC = 1.05 to 1.85) and hospitalized (RR = 1.57; 95% IC = 1.19 to 2.07) (Figure 2 and Figure 3). In the meta-regression, these results were less straightforward, as the predicted values (according to the test of moderators) were not statistically significant; although, the variable corresponding to ‘hospitalized’ was significant at a significance level of 0.1 according to a Wald test. The statistical heterogeneity found suggests that the observations were due to variations in the methodology applied in each of the included studies, and in their quality; these variations in combination with differences across populations and settings, and the type of predictors assessed (irrespective of whether included predictor was causal or not) among the included studies may lead to clinical heterogeneity. The Bayesian analysis confirmed the risks between the above-mentioned groups, though, subgroups analyses were limited by the studies available data for each group of analysis; hence, the findings should be interpreted with caution.

The findings also indicate that although there were differences statistically significant between the relative risks of dying in specific subgroups, the results might be a statistical artefact as it is possible that the populations’ characteristics (i.e., significant age differences, course and severity of the underlying disease, and coexistence of multimorbidity) were led by themselves to elevate the risk of mortality. Furthermore, the overall effect that hypervitaminosis B12 could have on mortality is more complex than a specific endpoint can capture, owing to the fact that all the included studies used as biomarker the total vB12 measurement; this test may mask a true B12 impairment or incorrectly suggest an elevation [1] since it does not measure the active form but the total of vB12. Likewise, the effect of vB12 assessed at a single moment in time on subsequent mortality (over a period of many years) is insufficient to confirm its prognostic value, as vB12 itself may change over time. In this respect, the fact that fifteen studies were based on national population health surveys evidenced a methodological error in prediction studies because these surveys provide up-to-date snapshots of the population’s health and behaviors. They determine only baseline exposures; they do not ascertain temporal outcomes, although they are linked to vital statistics and health care data. It is also important to note that the 28 studies included are based on available data given that the sample size was paid little attention, and that relevant predictors such as liver/kidney failure are not clearly defined and standardized among the studies.

All the prognostic studies published in this research topic fell under the umbrella of ‘exploratory’ and their results raised more questions than provided answers. In etiology and prediction research, the research questions, methods, and interpretations of results are fundamentally different [39]. The former explains whether an exposure is responsible for, or causes, an outcome–with adjustment of confounders, the latter uses multiple variables (not necessarily etiological causes) to predict the risk of future outcomes. Likewise, in clinical work, the role of uncertainty in decisions may play out differently if the decision refers to test causation, or to the probability/risk of an individual developing a particular outcome (i.e., death or complications).

### Comparison with Earlier Reviews

A 2024 systematic review [32] assessed six longitudinal studies encompassing 94,313 subjects to summarize the then current evidence on the associations of hypervitaminosis B12 and all-cause mortality risk. This review is constrained by limitations inherent in the available evidence, failing to pool data through meta-analysis, and concluding that evidence linking vB12 and mortality is inconsistent.

On the contrary, a 2024 dose-response meta-analysis [31] shows that hypervitaminosis B12 (≥542 pg/mL) is positively associated with all-cause mortality risk, particularly among older adults (aHR = 1.5; 95% CI 1.29 to 1.74, *I*^2^ = 81.8%; *n* = 10). These findings are based on 18 longitudinal studies comprising 53,468 subjects and 10,244 all-cause deaths. Nevertheless, sources of variation among study results (e.g., variety of different follow-up periods, distinct vB12 cut-off points, vB12 quantified at a single moment in time, among others) are not identified, nor is their impact on effect size quantified using meta-regression or an analysis of variance. In addition, vB12 concentrations are categorized inconsistently, which lead to misleading conclusions. Further, a statistically significant linear trend test does not indicate the presence of a monotonic relation [40].

Both reviews agree that the low quality of the literature and the significant heterogeneity found among the included studies creates difficulties for arriving at sound conclusions.

## 5. Limitations and Strengths

The scientific evidence on hypervitaminosis B12 as a predictor of mortality is hitherto weak, considering that the methodological quality of the evidence itself was low, and that the distinction between prediction and etiology [39] was not clearly stipulated, leaving the authors with studies that no longer answer a clear etiological or prediction question or may be misinterpreted.

All-cause mortality as outcome might generally be dominated by causes of death that are unrelated to the hypervitaminosis B12. Despite this, a direct assessment of all-cause mortality should not be ignored as it may indicate a possible exposure-outcome association. Nonetheless, the fact that total vB12 measurement was the test used (as a single indicator) to determine hypervitaminosis B12 may incorrectly suggest an excess of the active form of vB12.

It is also worth noting that this study included the use of frequentist and Bayesian statistical methods for data meta-analysis. They both helped to fill in the outline of prediction research of hypervitaminosis B12 with the best accuracy (with the available information). The comprehensiveness of searching and the representativeness of studies included are also important strengths as they provide insight on how the topic has been studied so far.

## 6. What Next?

The empirical lessons gained from this meta-analysis evidence the need for higher standards of empirical evidence to enable physicians to know whether hypervitaminosis B12 has or adds a predictive value on mortality or on a gloomy prognosis. This meta-analysis also evidences the need for studies that are powered for estimating overall mortality.

The findings of this meta-analysis generate new material for prediction model inquiries, the goal of which will be the overall predictive performance of the model, which should also be validated. The research questions and methods should be clear even if such studies intertwine prediction and causation to avoid biased estimates and erroneous conclusions.

## 7. Conclusions

At present, in observational medical research, the understanding of the relation of the hypervitaminosis B12 to overall mortality is mostly based on a causal interpretation of predictor effects, instead of alluding to its added value for risk prediction. This meta-analysis does not show that hypervitaminosis B12 represents a higher risk of all-cause mortality in adults.

## Figures and Tables

**Figure 1 nutrients-17-02184-f001:**
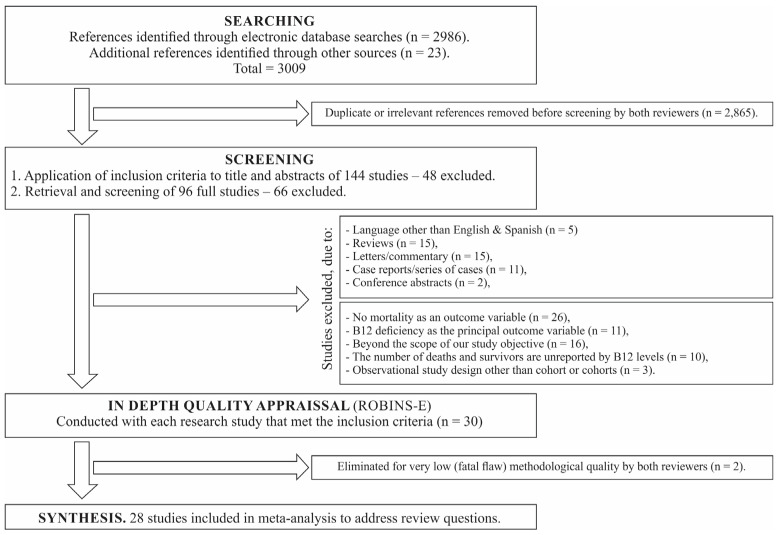
PRISMA flow diagram for identification of eligible studies.

**Figure 2 nutrients-17-02184-f002:**
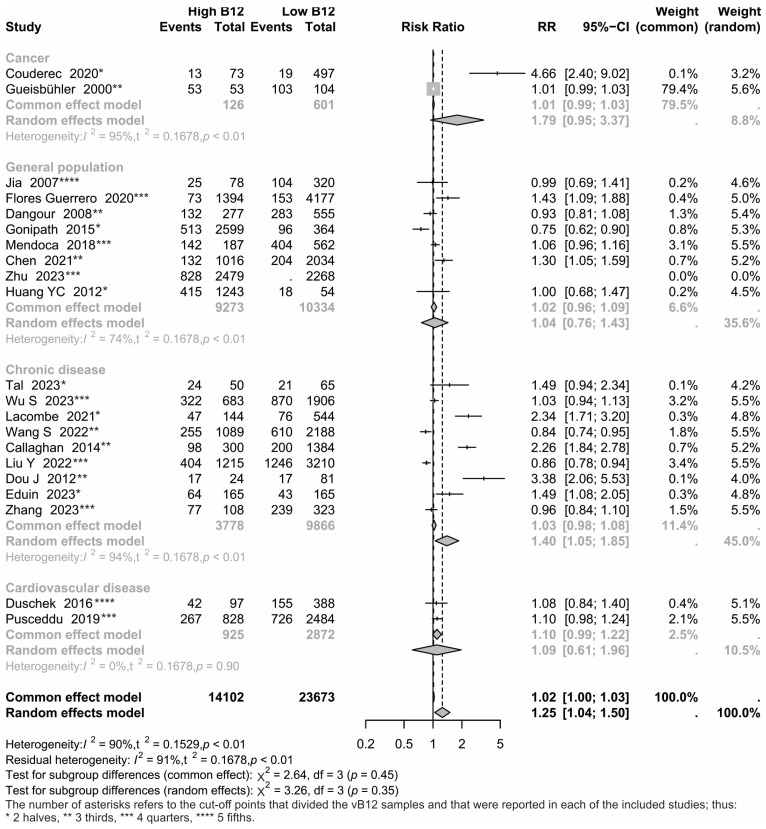
Forest plot and numerical results associated with the meta-analysis. Results are grouped by type of disease [3,4,5,6,7,8,9,10,11,13,17,19,20,21,24,25,26,27,28,29,30].

**Figure 3 nutrients-17-02184-f003:**
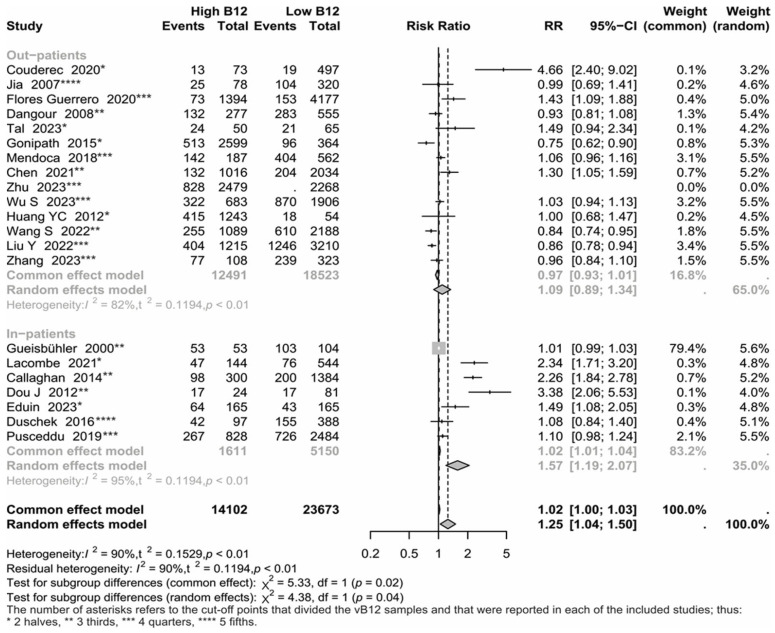
Forest plot and numerical results associated with the meta-analysis. Results are grouped by in-hospital-based studies versus outpatient-based studies [3,4,5,6,7,8,9,10,11,13,17,19,20,21,24,25,26,27,28,29,30].

**Figure 4 nutrients-17-02184-f004:**
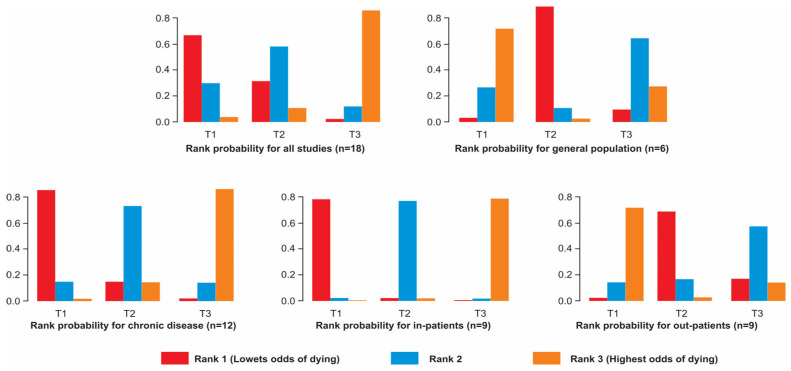
Rank probability diagrams associated with the network meta-analysis for all studies and by groups.

**Table 1 nutrients-17-02184-t001:** Meta-regression: inputs were studies based on subjects with chronic diseases, elderly population-based studies (yes or no), and in hospital (yes or no). Four studies with the largest publication bias were excluded.

	Estimate	S.E.	Z-Value	*p*-Value	95% Confidence Interval	
					Lower	Upper
Intercept	0.03	0.12	0.27	0.79	−0.2	0.26
Chronic diseases	−0.07	0.15	−0.44	0.66	−0.35	0.22
Elderly population	0.01	0.12	0.10	0.92	−0.23	0.26
In hospital	0.27	0.15	1.77	0.08	−0.03	0.58

## Data Availability

The template data collection forms, the data used for all analysis and any other materials used in the current meta-analysis are available from the corresponding author upon reasonable request.

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
