# Peer review of "The Controversial Issue of Hypervitaminosis B12 as Prognostic Factor of Mortality: Global Lessons from a Systematic Review and Meta-Analysis"

_nutrients, 2025, doi:10.3390/nu17132184_

Round 1

Reviewer 1 Report

Comments and Suggestions for Authors

Authors of a given meta-analysis try to clarify whether hypervitaminosis B12 can help predict mortality. Unfortunately, the article needs insensitive corrections:

  • Article needs to improve English.
  • Lines 54-65: too many ‘a’
  • In the results, it is worth focusing on the results significantly statistically.
  • What test was used to calculate the asymmetry?
  • In the case of heterogeneity, it is worth indicating its level.
  • Due to such high heterogeneity, can these results be generalized?
  • On figures 1-3 the title is given twice.
  • Question for figures 1 and 2: why are results given for common and random effect models? The authors describe in the results that the random model was chosen.
  • Question for figures 1 and 2: What do the '*,**,***,****' ' on the figures mean?
  • Why do the authors conclude that p=0.08 was still significant?
  • The discussion is very short and needs refinement.
  • Why the conclusion does not reflect the results: “High serum levels of B12 increased marginally the risk of all-cause mortality specifically among chronic diseases (RR = 1.40; 95% IC = 1.05 to 1.85) and hospitalized (RR = 1.57; 95% IC = 1.19 to 2.07)”?
Comments on the Quality of English Language

Article needs to improve English.

Reviewer 2 Report

Comments and Suggestions for Authors

Dear Authors,

Your manuscript about the review with the publication meta-analysis of hypervitaminosis B12 as prognostic factor of mortality, could be a interesting diagnostic tool for different diseases.

For this reason it is very necessary to improve the presentation of all figures of your manucript.

In particular is very recomendable to include a first Figure 1: With the Flow of the Selection: inclusion and exclution of the 28 included studies in the meta-analysis. And the presentation of the Tables and Graph is poor (there are double information; the figures caption appears twice) and all  Tables and Figures have to follow the instruction of the Journal: Nutrients.

All comment will improve the quality of you interesting manuscript of meta-analysis in B12 hipervitaminosis which a conclusion that is not as mortality factor.

Sincerely yours.

Comments on the Quality of English Language

No comments.

Reviewer 3 Report

Comments and Suggestions for Authors

This meta-analysis investigates whether hypervitaminosis B12 (serum levels >813 pg/mL) predicts all-cause mortality in adults. It synthesizes data from 28 longitudinal observational studies (69,610 participants, 15,815 deaths) using frequentist and Bayesian approaches. Results show a marginal increase in mortality risk among chronic disease patients (RR=1.40) and hospitalized individuals (RR=1.57), but meta-regression found no statistical significance. Bayesian analysis supported these subgroup risks but was limited by data availability. The authors attribute observed associations to methodological artefacts and clinical heterogeneity, concluding that hypervitaminosis B12 is not a robust prognostic marker for mortality

Overall, this is an interesting study that provides a nuanced analysis but would be improved by a requires stricter handling of heterogeneity and confounding factors to strengthen conclusions. In addition, methodological transparency and clinical contextualization would enhance utility.

Here is a summary of some issues:

Major Weaknesses

Inconsistent Handling of Heterogeneity

Section 3.3: High statistical heterogeneity (I²=89.7%) is noted but not sufficiently addressed. This will impact reliability and therefore, the authors should perform sensitivity analyses or subgroup analyses by study design/geography to identify sources of heterogeneity.

Overreliance on Single-Timepoint B12 Measurements

Section 3.1: 20/28 studies used a single vB12 measurement. Thereby, fluctuations in B12 levels over time are ignored, risking misclassification. The authors should improve this by excluding studies without repeated measurements or adjust for time-varying confounding.

Unclear Exclusion Criteria for "Fatal Flaws"

Section 2.2: Exclusion of studies with "fatal flaws" is undefined which makes it less transparent. Thereby, the authors should define specific methodological flaws (e.g., attrition bias, unadjusted confounders) in the protocol.

Minor Weaknesses

Incomplete Discussion of Confounding

Section 4: There is a lack of an approach to address comorbidities (e.g., liver/kidney disease) which independently affect B12 and mortality. The authors should stratify analyses by comorbidity severity or perform meta-regression adjusting for organ dysfunction.

Ambiguous Clinical Implications

Conclusions lack guidance for clinicians managing high B12 levels.

Potentially Inconsistent Terminology: "hypervitaminosis B12" is used throughout the manuscript while also "elevated B12" is used once, without defining thresholds. The authors should standardize terminology and align cutoffs with clinical guidelines (e.g., >900 pg/mL)

Comments on the Quality of English Language

accetable

Round 2

Reviewer 1 Report

Comments and Suggestions for Authors

Article has serious flaws. 

Author Response

No comments to work on

Reviewer 2 Report

Comments and Suggestions for Authors

No Comments.

Author Response

No comments

Reviewer 3 Report

Comments and Suggestions for Authors

Thank you for addressing the initial feedback and submitting your revised manuscript. While your revisions demonstrate engagement with key concerns, several critical issues remain unresolved which are listed below:

Major Unresolved Issues

Persistent Heterogeneity in Subgroup Analyses; Results section 3.3, Figure 1

  • The chronic disease subgroup shows extreme heterogeneity (I2=94%), yet no sensitivity analyses (e.g., leave-one-out) or meta-regression by geography/assay type were performed. This undermines confidence in the reported RR of 1.40. Without sensitivity analyses, heterogeneity sources remain unexplained. Important that you perform leave-one-out analyses and report variance explained by covariates (e.g., assay type, follow-up duration).

Single-Timepoint B12 Measurements; Methods section 2.1, Results  section 3.1

    • 20/28 studies relied on single B12 measurements, risking misclassification bias. Your response argued meta-analyses account for variability, but this does not mitigate time-varying confounding. Exclude studies without serial measurements or add sensitivity analyses excluding single-measurement studies.

Vague "Fatal Flaw" Criteria; Methods  section 2.7

    • While you clarified "fatal flaws" as high ROBINS-E risk, explicit thresholds (e.g., >15% attrition, unadjusted confounders) are still missing. Define exclusion thresholds using NHLBI guidelines (e.g., "high differential dropout" or "no adjustment for age/comorbidities"). see here: https://www.nhlbi.nih.gov/health-topics/study-quality-assessment-tools. It is possible that the supplementary files provide some clarification on this (can actually not open this file), but it should be clarified in the manuscript as well, if that is the case.

Incomplete Confounding Adjustment; Discussion section 4

    • Comorbidities were grouped broadly, but organ dysfunction (e.g., liver/kidney failure) was not stratified. The authors should stratify by organ dysfunction using available data or explicitly state this as a limitation.

I propose that major revision is still needed to address unresolved heterogeneity, measurement validity, and clinical applicability. While your work provides valuable insights, stricter methodological rigor is needed to solidify conclusions. I appreciate your efforts to improve the manuscript and remain available to review further revisions. Please ensure all changes are highlighted in the next submission for clarity.
